# Technical note: Entrainment-limited kinetics of bimolecular reactions in clouds

Christopher D. Holmes[1]

[1] Earth, Ocean, and Atmospheric Science, Florida State University, Tallahassee, FL 32306, USA

*Correspondence to*: Christopher D. Holmes (cdholmes@fsu.edu)

**Abstract.** The method of entrainment-limited kinetics enables atmospheric chemistry models that do not resolve clouds to simulate heterogeneous (surface and multiphase) cloud chemistry more accurately and efficiently than previous numerical methods. The method, which was previously described for reactions with first-order kinetics in clouds, incorporates cloud entrainment into the kinetic rate coefficient. This technical note shows how bimolecular reactions with second-order kinetics in clouds can also be treated with entrainment-limited kinetics, enabling efficient simulations of a wider range of cloud chemistry reactions. Accuracy is demonstrated using oxidation of $SO_2$ to $S(VI)$—a key step in formation of acid rain—as an example. Over a large range of reaction rates, cloud fractions, and initial reactant concentrations, the numerical errors in the entrainment-limited bimolecular reaction rates are typically $\ll 1\%$ and always $< 4\%$, which is far smaller than the errors found in several commonly used methods of simulating cloud chemistry with fractional cloud cover.

## 1 Introduction

Aqueous reactions in clouds play an important role in atmospheric chemistry, production of acid rain from $SO_2$ being a prominent example (Seinfeld and Pandis, 2016). Rapid heterogeneous (surface and multiphase) reactions can consume reactants within clouds, making the overall reaction rate dependent on entrainment to supply additional reactants from the surrounding air. Since clouds are sub-grid-scale features in many large-scale regional and global atmospheric models, accounting for these processes in chemical transport models is challenging. To address these challenges, Holmes et al. (2019) introduced entrainment-limited uptake, an algorithm to accurately and efficiently account for cloud chemistry occurring in just a fraction of a grid cell. The method incorporates cloud fraction and entrainment into

the kinetic rate expression, enabling calculation of concentrations in a partly cloudy model grid cell with very little computational effort. The original paper applied entrainment-limited uptake to first-order loss of nitrogen oxide compounds ($NO_2$, $NO_3$, $N_2O_5$) and showed that clouds are a globally significant sink for these gases (Holmes et al., 2019). The method has since been applied to nitrogen oxide isotopes (Alexander et al., 2020), nitrate in urban haze (Chen et al., 2021), dimethyl sulfide oxidation (Novak et al., 2021; Jernigan et al., 2022), and reactive halogens (Wang et al., 2021), all of which also involved first-order loss reactions in clouds. This note derives entrainment-limited reaction kinetics for bimolecular reactions with second-order kinetics so that the entrainment-limited method can be applied to a wider range of chemical systems that are important in the atmosphere.

## 2 Derivation

The computational challenge of cloud chemistry in a fractionally cloudy grid cell is that explicitly calculating reactant concentrations in the cloudy and clear fractions would increase the model's variables and computational effort. For cloud reactions with first-order kinetics, however, Holmes et al. (2019) showed that explicitly calculating concentrations within clouds can be avoided. For a reaction with loss frequency $k_i$ in clouds, the reaction rate in a partly cloudy grid cell is

$$R_1 = k_1 c \qquad \text{1a}$$
$$k_1 = k_i \left( \frac{x}{1+x} \right) \qquad \text{1b}$$

where $c$ is the reactant concentration in the grid cell (averaged over cloudy and clear fractions), $x/(1+x)$ is the fraction of reactant inside cloud, and

$$x = \frac{1}{2}(f' - k' - 1) + \frac{1}{2}(1 + k'^2 + f'^2 + 2k' + 2f' - 2k'f')^{1/2}, k' \equiv \frac{k_i}{k_c}, f' \equiv \frac{f_c}{1 - f_c}. \qquad 2$$

The cloud fraction is $f_c$ and $1/k_c$ is the mean residence time of air in clouds. The expression is exact for steady decay in which concentrations in and out of clouds decline at the same fractional rate. The overall idea is that kinetics governing grid-cell concentration follows the usual first-order form (Eq. 1a) with rate coefficients that depend on entrainment as well as chemical kinetics. We will follow a similar approach for bimolecular reactions.

Bimolecular reactions, $A + B \rightarrow$ products, typically follow second-order kinetic rate expressions of the form $R = k_{AB}c_A c_B$, where $k_{AB}$ is the rate coefficient. For reactions within clouds, the rate depends on gas-phase reactant concentrations within clouds, designated $c_{A,i}$ and $c_{B,i}$. These concentrations are related to the grid-average concentration via $c_{A,i}/c_A = x_A/f_c(1 + x_A)$, where $x_A$ is defined by Eq. 2 using the loss frequency for A within cloud. $c_{B,i}/c_B$ and $x_B$ are defined similarly. The loss frequency for A within cloud is the pseudo-first order rate $k_{A,i} = k_{AB}c_{B,i}$ and $k_{B,i} = k_{AB}c_{A,i}$ is the analogous loss for B. This forms a system of equations that collectively define gas-phase, in-cloud reaction rates for bimolecular reactions:

$$k_{A,i} = k_{AB}c_B \left( \frac{x_B}{f_c(1 + x_B)} \right) \tag{3a}$$

$$k_{B,i} = k_{AB}c_B \left( \frac{x_A}{f_c(1 + x_A)} \right) \tag{3b}$$

The system of equations 2 and 3 can be solved by root finding methods or fixed-point iteration. After evaluating $x_A$ and $x_B$, the overall reaction rate in a partly cloudy grid cell is found by substituting Eq. 3a into Eq. 1:

$$R_2 = k_2 c_A c_B \tag{4a}$$

$$k_2 = \frac{k_{AB} x_A x_B}{f_c(1 + x_A)(1 + x_B)}. \tag{4b}$$

Equation 4b is the exact form of the entrainment-limited bimolecular reaction rate coefficient. The grid-cell concentrations $c_A$ and $c_B$ typically have units molecule $cm^{-3}$ and the bimolecular rate coefficients $k_2$ and $k_{AB}$ typically have units $cm^{-3}$ molecule$^{-1}$ s$^{-1}$.

We can also derive an approximation to the entrainment-limited bimolecular rate coefficient that does not require iteration to solve. In the limit where the in-cloud reaction is much faster than entrainment ($k_{A,i} \gg k_c$ or $k_{B,i} \gg k_c$), the grid-scale losses of A and B are determined by the rate at which the limiting reactant is entrained into clouds:

$$R_2 \approx f'k_c \min(c_A, c_B). \tag{5}$$

In the limit where in-cloud reactions are slow ( $k_{A,i} \ll k_c$ and $k_{B,i} \ll k_c$ ) or the cloud fraction approaches 1, the losses follow second-order kinetics determined by the grid-scale mean concentrations:

$$R_2 \approx f_c k_{AB} c_A c_B. \qquad\qquad 6$$

80 Combining these limits gives an approximation of the entrainment-limited bimolecular loss rates, expressed as a grid-scale 2$^{\text{nd}}$ order rate coefficient

$$k_2 \approx \left( \left( \frac{f' k_c \min(c_A, c_B)}{c_A c_B} \right)^{-1} + (f_c k_{AB})^{-1} \right)^{-1}. \qquad\qquad 7a$$

Although Eq. 7a is finite and well defined for all values of $f_c$, numerical overflow could occur with finite-precision arithmetic when $f_c$ approaches 0 or 1. To improve stability and accuracy, numerical
85 calculations can use the equivalent expression

$$k_2 \approx \frac{f_c k_c k_{AB} \min(c_A, c_B)}{k_c \min(c_A, c_B) + \left(1 - f_c\right) k_{AB} c_A c_B}. \qquad\qquad 7b$$

This approximate entrainment-limited bimolecular reaction rate coefficient (7a or 7b) can be used in Eq. 4a.

## 3 Evaluation

90

The accuracy of entrainment-limited bimolecular reaction rates will now be demonstrated using oxidation of S(IV) by aqueous H$_2$O$_2$, which is a prominent step in the formation of S(VI) and acid rain, as an example (Chameides, 1984). One key aqueous reaction is $HSO_3^- + H_2O_2 + H^+ \rightarrow SO_4^{2-} + 2H^+ + H_2O$, where the reactants are dissolved forms of gaseous SO$_2$ and H$_2$O$_2$. While the reaction occurs in
95 cloud droplets, the reaction rate can be expressed in terms of the gas-phase concentrations of SO$_2$ and H$_2$O$_2$ by incorporating the solubility and dissociation equilibria, cloud liquid water content, and aqueous kinetics into the effective, gas-phase rate coefficient (e.g., Park et al., 2004). For a cloud with 1 g m$^{-3}$ liquid water at pH 5, 284 K, and 800 hPa, the effective, gas-phase bimolecular rate coefficient is $k_{\text{eff}} = 3.7 \times 10^{-14}$ cm$^3$ molecule$^{-1}$ s$^{-1}$, which will be used in examples below. A similar approach can be
100 applied to other bimolecular aqueous reactions.

Figure 1 shows that the exact entrainment-limited algorithm (Eq. 4) is nearly identical to a reference solution in a two-box model that explicitly represents concentrations inside clouds and entrainment mixing with clear air. The approximate entrainment-limited solution (Eq. 7) also resembles the exact entrainment-limited and reference solutions, but remaining reactant concentrations diverge by 3 % after 1 hour and 10 % after 4 hours. Two other cloud chemistry methods that are used in current atmospheric chemistry models are also shown in Figure 1: the thin-cloud approximation, in which loss is computed for the entire grid cell using grid-average liquid water content, and the cloud partitioning method, in which only reactants within the cloudy fraction can react, but the concentrations are homogenized across cloudy and clear regions each time step of the chemical solver. Holmes et al. (2019) describe these other methods in greater detail. Both of the other methods diverge from the reference solution and entrainment-limited method by large amounts.

Figure 2 shows accumulated error in the entrainment-limited kinetics over a wide range of initial reactant concentrations and cloud fractions. Results are presented as the error in total product formed, relative to the reference two-box model, after one hour of integration. Over most of the parameter space, the errors in the entrainment-limited calculations are much less than 1 %. The largest errors occur over a narrow range of $k_{AB}c_B/k_c$ values in regions that are about half cloudy and these errors do not exceed 4 %. By the same metric, the approximate entrainment-limited bimolecular algorithm has up to 10-30 % error (Figure 2). The thin-cloud method has much larger errors than either of the entrainment-limited methods over most of the parameter space in Figure 2. These thin-cloud errors exceed 1000 % when cloud fractions are small and in-cloud reactions are fast. As $f_c$ approaches 1, however, the thin-cloud method has increasingly good accuracy, with errors under 0.1 % for $f_c \geq 0.97$. Numerical codes can, therefore, use thin-cloud instead of entrainment-limited kinetics when $f_c \gtrsim 0.97$ for computational efficiency.

The relative computational performance of these cloud chemistry methods depends on numerous factors, such as reactant concentrations, cloud fraction, differential equation solver, error tolerances, optimizations, programming language, etc. Some general comparisons can be made, however, using the

conditions of Figure 1. (Code for timing tests is provided in the supplement.) When evaluating the instantaneous reaction rate (e.g. at time $t = 0$ in Fig. 1), the approximate entrainment-limited method is about 15 times faster than the exact method and the thin-cloud method is about 100 times faster. There is much less disparity in execution times when integrating the solution over time, however, because numerical solvers have many additional components. For the integration shown in Figure 1, the approximate entrainment-limited method is about 2.3 times faster than the exact method. The thin-cloud method, meanwhile, is only about 25 % faster than the exact entrainment-limited solution, because the solver takes many more internal time steps as concentrations quickly decline. Speed differences between the methods would likely diminish further in a chemical mechanism with more compounds and reactions. Nevertheless, this comparison shows that computational speed should not be a major impediment to adopting entrainment-limited reaction kinetics.

The entrainment-limited approach is best suited for applications and models that do not require highly detailed cloud and aqueous chemistry. For example, the derivation above assumes that reactants A and B are consumed in only one reaction. While additional in-cloud reactions and reactants can be incorporated into the pseudo-first order loss rates (Eq. 3), to account for their effects on $x_A$ and $x_B$, solving the system becomes more computationally intensive as more reactants are involved. For cloud reactions that depend on [H$^+$], the pH must be assumed or calculated via another method because it is infeasible to account for the relevant aqueous equilibria within the entrainment-limited equations. Overcoming these limitations, however, requires explicit representation of reactant concentrations and entrainment in the cloudy fraction of a grid cell, along with the extra computational burden that incurs. Despite the progression of atmospheric models to ever higher resolutions, fractional cloudiness is likely to remain a feature of many global and regional models for many years to come, necessitating some means of accounting for its effect on chemistry.

# 4 Conclusion

The results here and in the earlier work of Holmes et al. (2019) show that the entrainment-limited reaction kinetics can provide an efficient and accurate means of representing heterogeneous cloud

chemistry in atmospheric models with fractional cloud cover. By incorporating cloud fraction and entrainment into the rate coefficient, the usual first- and second-order rate expressions are retained, allowing the entrainment-limited kinetics to be easily implemented in numerical codes. The entrainment-limited approach provides far greater accuracy than other methods currently in use; typical errors for bimolecular reactions are << 1 % error after 1 hour and always < 4 %. Entrainment-limited kinetics have already been applied to numerous first-order reactions and the extension here to bimolecular reactions should further expand its applicability and usefulness in atmospheric chemistry modeling.

## Code availability

Python code implementing the entrainment-limited bimolecular kinetics is provided in the supplement.

## Competing interests

The author declares that he has no conflicts of interest.

## Acknowledgments

I thank Daniel Jacob and Mike Long for helpful discussions. This work was supported by the NASA New Investigator Program (grant NNX16AI57G).

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

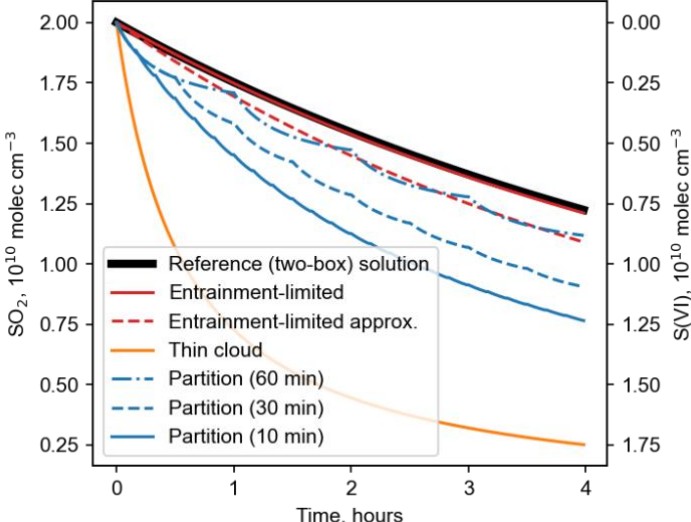

215

**Figure 1: Comparison of numerical solutions for reaction of dissolved SO$_2$ with H$_2$O$_2$ in cloud water in a partly cloudy region. Calculations use conditions $T$ = 284 K, $p$ = 800 hPa, 1 g m$^{-3}$ liquid water in cloud, pH = 5, $f_c$ = 0.2, $k_c$ = 1 h$^{-1}$, $k_{eff}$ = 3.7 $\times$ 10$^{-14}$ cm$^3$ molecule$^{-1}$ s$^{-1}$, and initial concentrations $c_{SO_2}$ = $c_{H_2O_2}$ = 2.0 $\times$ 10$^{10}$ molecule cm$^{-3}$ (1 ppb). For the cloud partitioning method, the numbers in parentheses give the**

220 **time step for homogenizing reactant concentrations.**

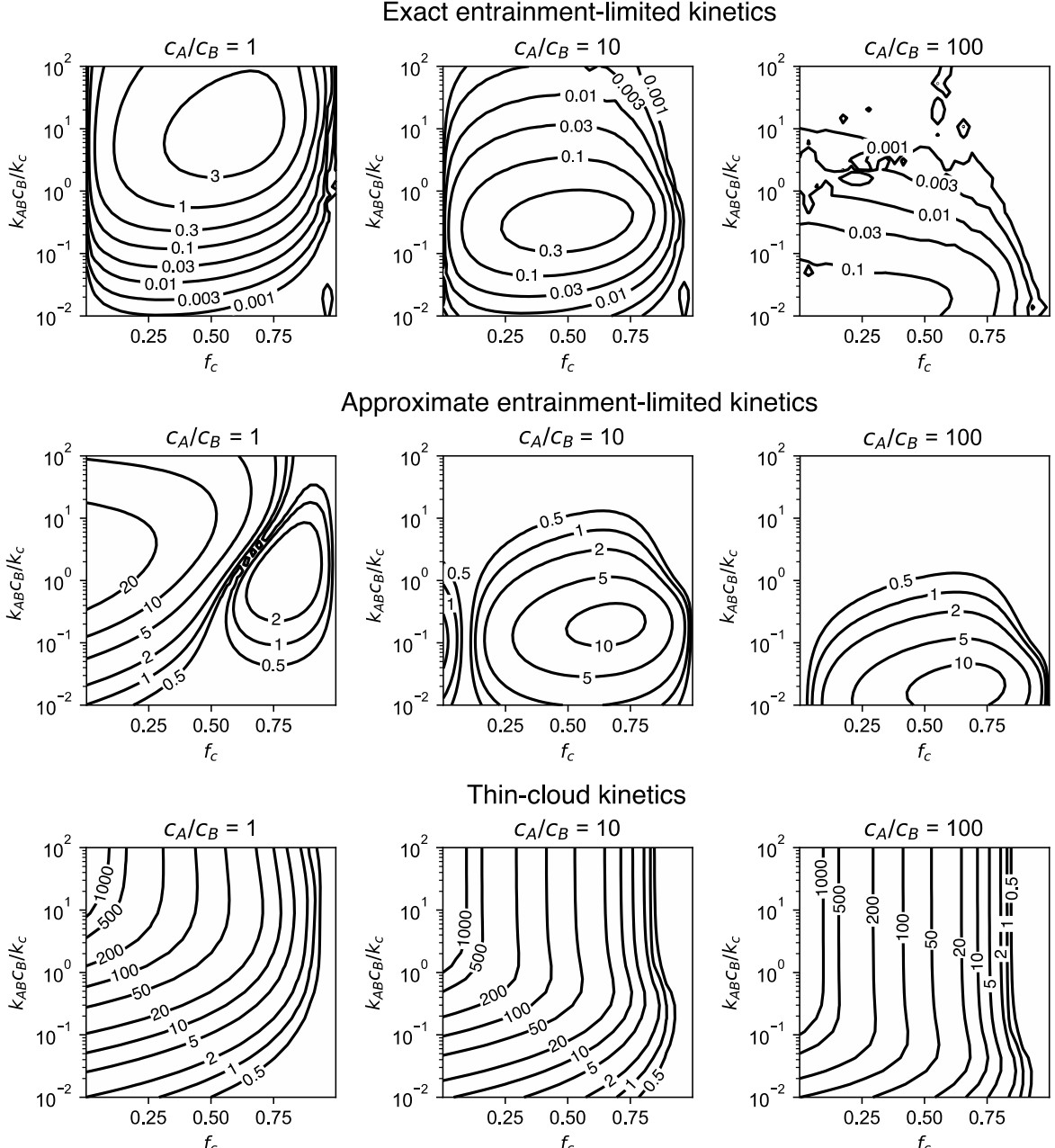

**Figure 2: Accuracy of exact entrainment-limited bimolecular kinetics (Eq. 4, top row), approximate entrainment-limited kinetics (Eqs. 4a and 7, middle row), and thin-cloud kinetics (bottom row). Accuracy is shown as the percent difference (%) in the cumulative loss of reactants after 1 hour relative to a reference two-box model. For each panel, calculations are performed for a grid of 30×30 points linearly distributed over $f_c \in [0.001, 0.999]$ and logarithmically distributed over $k_{AB}c_B/k_c \in [0.01, 100]$.**