# Peer review of "Technical note: Entrainment-limited kinetics of bimolecular reactions in clouds"

_Atmospheric Chemistry and Physics, 2021_

## Referee Comment (RC2)

Review of Holmes ACPD

Holmes provides a method to account for entrainment-limited kinetics in second-order aqueous-phase reactions in clouds in large-scale models of the atmosphere that do not resolve clouds. This builds upon is prior work in developing a method for first-order reactions. He uses the example of an important second-order aqueous-phase reaction in the atmosphere, oxidation of S(IV) by H2O2. He quantifies the numerical errors in this method by comparing it with results from a two-box model that explicitly represents clouds and finds that the errors are relatively small (typically << 1%).

This paper is an important advance, and I am left very curious how this will impact sulfate formation, and other important aqueous-phase reactions, in large scale models of atmospheric chemistry. I hope a paper on this is forthcoming. My suggestion for improvement of this paper is to be clear on whether the concentrations and rate constants are for the gas-phase or the aqueous-phase. In Holmes et al., GRL, 2019, this was clear, but it is not clear here. This could be done by explicitly stating this in the text, as was done in Holmes et al., GRL, 2019, and/or by providing units for each variable. On a related topic, I wonder why the terminology has changed from the 2019 paper. In the 2019 paper, the variable $c$ was used for gas-phase concentrations. In this paper, [A] and [B] are used instead. Often, square brackets signify aqueous-phase concentrations. Perhaps use $c_A$ and $c_B$ for consistency?

---

## Author Response (AR1)

**I appreciate the thoughtful and supportive comments from both reviewers. I have made changes to the manuscript to answer all of the questions that the reviewers raised. The changes and my replies to reviewers are in bold text below.**

**These responses have already been posted in Author Comments in the public discussion.**

**Comments from Reviewer 1**

The author has extended an innovative and efficient technique to deal with the "partial cloud" issue, allowing modelers to simulate bimolecular reactions limited by the rate of entrainment. The results suggest that the technique can provide greater accuracy than the current standard, with little implied additional cost.

The central question is both interesting and important, providing an incremental step towards resolving the problem of how to deal with "partial cloudiness" in atmospheric chemistry. The results generally support the conclusions given, although I would like to see some additional information on performance. Although the note does not provide a major advance, the author recognizes the limitations of the technique and does not over-sell the findings. I particularly appreciated the inclusion of Python code to implement the model.

The note is well-structured and compelling. I believe that, with only minor revisions, it is appropriate for publication in ACP. However, I have made some suggestions (and identified one typo) below which I believe could improve the paper.

*Major comments*

All atmospheric chemistry-transport modeling involves a tradeoff of computational resources against accuracy, so it would be useful if the author provided a sense of what (if any) additional computational burden is caused by the implementation of the exact and approximate entrainment-limited approaches, as well as the thin-cloud or partitioning approaches. This would allow assessment of the true advantage of using this method.

**I will update the sample Python program to include some basic timing tests comparing the methods. I will add the following paragraph to Section 3.**

**"The relative computational performance of these cloud chemistry methods depends on numerous factors, such as reactant concentrations, cloud fraction, differential equation solver, error tolerances, optimizations, programming language, etc. Some general comparisons can be made, however, using the conditions of Figure 1. (Python code for timing tests with an implicit Radau solver is provided in the supplement.) When evaluating the instantaneous reaction rate (e.g. at time $t = 0$ in Fig. 1), the approximate entrainment-limited method is about 15 times faster than the exact method and the thin-cloud method is about 100 times faster. There is much less disparity in execution times when integrating the solution over time, however, because numerical solvers have many additional components. For the integration shown in Figure 1, the approximate entrainment-limited method is about 2.3 times faster than the exact method. The thin-cloud method, meanwhile, is only about 25 % faster than the exact entrainment-limited solution, because the solver takes many more internal time steps as concentrations quickly decline. Speed differences between the methods would likely diminish further in**

**a chemical mechanism with more compounds and reactions. Nevertheless, this comparison shows that computational speed should not be a major impediment to adopting entrainment-limited reaction kinetics."**

Figure 2 is enlightening, but it would be most helpful if some additional information could be given about how consistent the thin-cloud error is. I would suggest including (around line 112) both the minimum and maximum error of the thin-cloud approximation over the domain; and, if it is ever within 30%, a third row could be included on Figure 2 which shows the error due to the "thin-cloud" approximation. Although the additional three panels are presumably not particularly data-rich, this would help to show whether the benefits inferred from Figure 1 are consistent across all conditions.

**I will add the suggested panels showing the thin cloud method to Figure 2 and a sentence describing the results: "The thin cloud method has much larger errors than either of the entrainment-limited methods over most of the parameter space in Figure 2. These thin-cloud errors exceed 1000 % when cloud fractions are small and in-cloud reactions are fast. As $f_c$ approaches 1, however, the thin-cloud method has increasingly good accuracy, with errors under 0.1 % for $f_c$ > 0.97. Numerical codes could, therefore, use thin-cloud instead of entrainment-limited kinetics when $f_c$ >~ 0.97 for computational efficiency."**

The nature of the equations (especially 5 and 7) is concerning with regards to the behavior in the limits of $f_c$ -> 0 and $f_c$ -> 1. In exploring the parameter space for Figure 2, how many points were used? And how were the limits handled? How well behaved is the solution for values of fc which are almost, but not quite, equal to 0 or 1, and how important are these limits? The supplemental material indicates that a there are transitions at $f_c = 10^{-4}$ and $f_c = 0.99$; are these continuous in value? Some discussion of this behavior would be helpful.

**Equations 5 and 7 (approximate entrainment-limited kinetics) are well defined in both limits $f_c$ -> 0 and $f_c$ -> 1. I will add two sentences to make this clear:**
**"Although Eq. 7a is finite and well defined for all values of $f_c$, numerical overflow could occur with finite-precision arithmetic when $f_c$ approaches 0 or 1. To improve stability and accuracy, numerical calculations can use the equivalent expression**

$$k_2 \approx \frac{f_c k_c k_{AB} \min(c_A, c_B)}{k_c \min(c_A, c_B) + (1 - f_c) k_{AB} c_A c_B}.$$ **7b**

**I will expand the caption of Figure 2 to state how many points were used. "For each panel, calculations are performed for a grid of 20×20 points linearly distributed over $f_c \in$ [0.001, 0.999] and logarithmically distributed over $k_{AB} c_B / k_c \in$ [0.01, 100]."**

**The transition at $f_c$ > 0.99 is essentially continuous. As stated above, the code has errors < 0.1 % after 1 hour under these conditions. The transition at $f_c < 10^{-4}$ is not continuous, but it is meant to be a suggestion that some users might choose to ignore cloud chemistry for computational speed when cloud fractions are exceedingly small. I will clarify this in the supplemental code comments.**

*Minor comments*

Line 42: Typo – "ins" should be "in"

**Done**

**Comments from Reviewer 2**

Holmes provides a method to account for entrainment-limited kinetics in second-order aqueous-phase reactions in clouds in large-scale models of the atmosphere that do not resolve clouds. This builds upon is prior work in developing a method for first-order reactions. He uses the example of an important second-order aqueous-phase reaction in the atmosphere, oxidation of S(IV) by H2O2. He quantifies the numerical errors in this method by comparing it with results from a two-box model that explicitly represents clouds and finds that the errors are relatively small (typically << 1%).

This paper is an important advance, and I am left very curious how this will impact sulfate formation, and other important aqueous-phase reactions, in large scale models of atmospheric chemistry. I hope a paper on this is forthcoming. My suggestion for improvement of this paper is to be clear on whether the concentrations and rate constants are for the gas-phase or the aqueous-phase. In Holmes et al., GRL, 2019, this was clear, but it is not clear here. This could be done by explicitly stating this in the text, as was done in Holmes et al., GRL, 2019, and/or by providing units for each variable. On a related topic, I wonder why the terminology has changed from the 2019 paper. In the 2019 paper, the variable $c$ was used for gas-phase concentrations. In this paper, [A] and [B] are used instead. Often, square brackets signify aqueous-phase concentrations. Perhaps use $cA$ and $cB$ for consistency?

**Thank you for the supportive comments and constructive suggestions clarity. I will adopt the notation with $c_A$ and $c_B$ for gas-phase concentrations. I will also add the phrase "gas-phase" when these variables are introduced and a sentence about units, both in Section 2. As the reviewer suggests, a future paper will address the sulfate formation comprehensively.**